# Electrochemical Analysis of Free Glycerol in Biodiesel Using Reduced Graphene Oxide and Gold/Palladium Core-Shell Nanoparticles Modified Glassy Carbon Electrode

Victor Magno Paiva [1], Kelly Leite dos Santos Castro Assis [2], Braulio Soares Archanjo [2], Daniela Ramos Ferreira [1], Carlos Alberto Senna [2], Emerson Schwingel Ribeiro [1], Carlos Alberto Achete [2] and Eliane D'Elia [1,*]

[1] Institute of Chemistry, Federal University of Rio de Janeiro, University City, Rio de Janeiro 21941-909, Brazil; victor.magno13@gmail.com (V.M.P.); rfdaniela@hotmail.com (D.R.F.); emersonsr@iq.ufrj.br (E.S.R.)

[2] National Institute of Metrology, Quality and Technology, Duque de Caxias 25250-020, Brazil; klcassis@gmail.com (K.L.d.S.C.A.); bsarchanjo@inmetro.gov.br (B.S.A.); carlosalbertosenna15950@gmail.com (C.A.S.); caachete@inmetro.gov.br (C.A.A.)

\* Correspondence: eliane@iq.ufrj.br

**Abstract:** Glycerol is a major byproduct obtained in the production of biodiesel, an important renewable fuel. The presence of free glycerol in fuel can have structural and performance consequences with respect to the engine, making fuel quality control important. The standard method to analyze glycerol in biodiesel is gas chromatography, a time-consuming and expensive technique. In this context, an electrode based on glassy carbon electrodes (GCEs) modified with reduced graphene oxide and core-shell gold@palladium nanoparticles was developed for the determination of glycerol in biodiesel. The free glycerol analysis was performed in the aqueous phase obtained by liquid–liquid extraction from a biodiesel sample. Cyclic voltammetry was chosen as the method for glycerol electrochemical analysis to regenerate active sites and promote greater sensor stability. The modified Au@Pd/rGO/GCE electrode showed an excellent performance, obtaining a linear range of 18.2 to 109 $\mu mol\ L^{-1}$ with a correlation coefficient of 0.9895, limits of detection and quantification of 5.33 and 17.6 $\mu mol\ L^{-1}$, respectively, high stability during 1000 cycles, and recovery values of 86% and 87% in the quantification of glycerol in biodiesel samples. The proposed method proved to be a great alternative for the analysis of glycerol in biodiesel, being a fast, sensitive, and low-cost technique due to its high stability and the use of small quantities of reagents.

**Keywords:** glycerol; core-shell gold@palladium nanoparticles; reduced graphene oxide; cyclic voltammetry





## 1. Introduction

Glycerol is a very important non-toxic substance that is often used in various industries, including those of cosmetics, foods, pharmaceuticals, and personal care products. It has recently been used as an alternative energy source through the oxidation of alcohol in fuel cells [1–4]. It is also important to note that glycerol is obtained as a co-product in biodiesel production, an important alternative fuel that reduces dependence on oil and meets the green demand to avoid polluting the environment. For this reason, the use and study of biodiesel have increased greatly in recent years [5].

Biodiesel is produced by transesterification of vegetable oils, animal fats, or waste cooking oils in the presence of a short-chain alcohol (methanol or ethanol) and a strong base catalyst [6]. In this process glycerol is obtained as a residual product, where its quantity and purity depend on the conversion method, the nature of the alcohol, and the catalysts used [7,8]. The presence of glycerol can result in structural and engine performance consequences, including storage problems, clogging of injector nozzles, and the burning of glycerol that can release acrolein, a toxic substance which is highly irritating to mucous

membranes, skin, eyes, and respiratory system and contributes to respiratory diseases as respiratory distress and noncardiogenic pulmonary edema, especially in large urban centers. Thus, strict control is required for the presence of alcohol in biodiesel, and it is regulated in several areas, including the United States (ASTM D6751), Germany (DINV 51606), Europe (EN 14214), and China (GB/T20828-2007) [5]

It is widely produced by the transesterification of vegetable oil or animal fat, with methanol or ethanol and a strong base as catalysts.

There are several methods for the analysis of free glycerol. Among these are chromatographic (the standard method ASTM D6584 [9]), amperometric, spectrophotometric, and enzymatic [7,8,10–14] methods. However, the most widely and currently studied method is electrochemical, due to its analysis being less laborious, less expensive, and at the same time sensitive and fast. Some electrodes for the quantification of glycerol have already been reported, among them the Cu/CuO electrode [8,15], the Au(111)/SiO$_2$ cavity/ITO electrode [5], electrodes based on nickel [2,16], and the PtRu electrode [17].

Palladium has been widely used for oxidizing alcohols due to its relatively low cost, abundance, and high catalytic activity. However, the advancement of science allowed the combination of electrocatalytic materials to obtain an improvement in the performance of the electrode. The synergism resulting from the combination of metals provided an increase in the efficiency of the catalytic of metals [18,19], while the use of graphene-based materials as a support for deposition of metallic nanoparticles provided a gain in surface area, generating dense active sites and increasing electrical conductivity, durability, and mechanical stability [20–22].

Thus, catalytic bimetallic systems based on palladium with other transition metals such as Au, Cu, Co, Ag, and Ni have been the target of many researchers; however, the positive synergistic effect between Au and Pd has shown to be catalytic for a greater oxidation of alcohols. In addition, gold is able to oxidize the CO intermediates that bind to the active site of the Pd, increasing the catalytic activity of palladium and its tolerance to poisoning, and improving the efficiency of the electrode [23–25].

This work aims to develop an electrochemical sensor for the quantification of free glycerol in biodiesel from the modification of the surface of the glassy carbon electrode with reduced graphene oxide and gold and palladium nanoparticles (Au@Pd/rGO/GCE). First, due to the presence of impurities such as methanol, catalysts, water, and soap in the glycerol from biodiesel, purification needed be performed before analysis using liquid–liquid extraction.

## 2. Experimental

### 2.1. Reagents

Glycerol (99.5% *v/v*) and potassium hydroxide (85.0% *w/w*) were purchased from Isofar (São Paulo, Brazil); chloroauric acid (99.5% *w/v*), palladium chloride (99.5% *w/w*) sulfuric acid (95–98.0% *v/v*), sodium chloride (99.0% *w/w*), hydrogen phosphate disodium (99.0% *w/w*), and potassium hydrogen phosphate (99.0% *w/w*) were bought from Merck (Rio de Janeiro, RJ, Brazil); and expanded graphite was purchased from Nacional de Grafite (São Paulo, SP, Brazil). All solutions were prepared using ultrapure water obtained from the Millipore Gradient A10 system (Watertown, MA, USA), with resistivity of 18.2 M$\Omega$ cm at 25 °C.

### 2.2. Electrochemical Apparatus

The electrochemical tests were performed by the Autolab PGSTAT204 potentiostat (Metrohm-AUTOLAB, Utrecht, the Netherlands). A 3-electrode system was used for analysis, where the modified glassy carbon electrodes (GCE, 3.14 mm$^2$ area, Lab solutions, São Paulo, Brazil) were used as the working electrode, the platinum wire (99.9%) as an auxiliary electrode, and an Ag|AgCl|KCl(sat.) (3.0 mol L$^{-1}$ KCl) as a reference electrode.

### 2.3. Synthesis of the Reduced Graphene Oxide

Before the modification of the GCE with reduced graphene oxide (rGO), the GCE was cleaned according to the following steps. First, the electrode was polished with alumina powder suspension (0.30 µm), followed by sonication in 3.0 mmol L$^{-1}$ of HNO3 for 10 min. It was submitted to electrochemical cleaning with 0.50 mol L$^{-1}$ sulfuric acid using the cyclic voltammetry technique at a scan rate of 1.0 V s$^{-1}$ in a potential range of −1.0 to 1.0 V.

After this, the reduced graphene oxide (rGO) was produced from graphite oxide synthesized by the method of Hummers and Offman [26] according to previous works [27]. For this, 5 µL of a 1.0 mg mL$^{-1}$ solution previously exfoliated for 20 h at a frequency of 40 kHz was applied to the surface of the GCE; once dried, the electrode was subjected to electrochemical reduction by chronoamperometry under −1.5 V for 60 s in phosphate buffer at pH 7.4.

### 2.4. Modification of Glassy Carbon Electrodes with Core-Shell Au@Pd Nanoparticles (Au@Pd/rGO/GCE)

After the modification of GCE with rGO (rGO/GCE) the electrode was subjected to cyclic voltammetry for 500 cycles, with a 5.0 V s$^{-1}$ scan rate in the range of −1.0 to 1.0 V using a solution of H[AuCl$_4$] and PdCl$_2$, with a 1:5 (0.65 and 3.25 mmol L$^{-1}$, respectively) stoichiometric ratio, in 0.50 mol L$^{-1}$ H$_2$SO$_4$. Finally, for activation, the electrode was subjected to cyclic voltammetry in a KOH 0.030 mol L$^{11}$ solution for 5 cycles, with a 0.50 V s$^{-1}$ scan rate in the potential range of −0.80 to 0.80 V.

### 2.5. Characterization of Graphene-Based Material

The graphene oxide used in this work has previously been studied and characterized. This can be seen in our previous article [27]. The graphene-based materials were characterized by scanning electron microscopy, X-ray diffraction, Raman Spectroscopy, X-ray photoelectron spectrometry, and atomic force microscopy.

The Randles–Sevcik equation (Equation (1)) was used to calculate the electroactive area [28–30]. The study was performed using the cyclic voltammetry technique in 20 mmol L$^{-1}$ K$_3$[Fe(CN)$_6$] in 3.0 mol L$^{-1}$ KCl at different scan rates.

$$i_{p,a} = 2.69 \times 10^5 \, n^{\frac{3}{2}} \, A \, C \, D^{\frac{1}{2}} \, v^{\frac{1}{2}} \tag{1}$$

where $i_{p,a}$ is the anodic peak current (A), $A$ is the electroactive surface area (cm$^2$), $n$ is the number of electrons transferred in the process, $C$ is the [Fe(CN)$_6$]$^{3-}$ concentration (mol cm$^{-3}$), $D$ is the [Fe(CN)$_6$]$^{3-}$ diffusion coefficient ($6.2 \times 10^{-6}$ cm$^2$ s$^{-1}$), and $v$ (V s$^{-1}$) is the rate scan.

The heterogeneous electron transfer (HET) rate constant ($k^0$) of the electrode with graphene oxide and reduce graphene oxide was calculated the $k^0$ according to the Nicholson equation [31–33] (Equation (2)).

$$\Psi \sqrt{\frac{\Pi D_o n F v}{RT}} = k^0 \tag{2}$$

where $R$ is the gas constant, $\alpha$ is the transfer coefficient, $n$ is the number of electrons transferred, $v$ is the scan rate (V s$^{-1}$), $F$ is the Faraday constant, $T$ (K) is the temperature of the system, $\Psi$ is a dimensionless charge transfer, and $D_0$ is the oxidation diffusion coefficient.

### 2.6. Characterization of the Electrode Surface

The evaluation of morphological characteristics such as the distribution, form, and size of particles on the surface of the electrodes was studied by scanning electron microscopy (SEM) in a FEI Helios Nanolab 650 apparatus working at 10 or 20 kV. TEM analysis was performed in a probe-corrected FEI Titan 80–300 apparatus; the images were collected in scanning transmission electron microscopy (STEM) mode using a high-angle annular dark

field (HAADF) detector. An X-ray energy dispersive spectroscopy (EDS) detector was used in elemental analysis. For TEM analyses the electrode was scraped in paper, then sonicated in ethanol, and finally a single drop was placed in a conventional TEM copper grid with a thin holey carbon film.

*2.7. Partial Validation*

One of the requirements when developing a method is to guarantee that it is effective for what is proposed, and thus after evaluating among the proposed electrodes the electrode with the greatest sensitivity, it was submitted for analytical validation according to the criteria of the Brazilian standard determined by INMETRO, where the linearity, limit of detection (LOD), limit of quantification (LOQ), precision (repeatability and intermediate precision), and accuracy were studied [34].

### 2.7.1. Linearity

Linearity was evaluated in terms of the correlation coefficient, homoscedasticity, and normality of the residues, studied using Pearson's coefficient, Cochran's test, and the Anderson–Darling test, respectively. For the correlation coefficient, r was used as a parameter (r > 0.9000) [35], while for the Cochran [36] and Anderson–Darling [37] tests, the *p*-values were used to assess the null hypotheses of the residual tests. The linear curve, in the linear range from 18.2 to 109 µmol L$^{-1}$, was generated from a 1.37 mol L$^{-1}$ glycerol stock solution, where aliquots were added to the electrochemical cell containing 15 mL of 0.030 mol L$^{-1}$ KOH. All concentrations were analyzed in triplicate.

### 2.7.2. Limit of Detection and Quantification (LOD e LOQ)

The detection (LOD) and quantification limits (LOQ) were obtained from the slope ($\alpha$) and the intercept standard deviation ($\partial$) of the analytical curve using Equations (3) and (4) [35]:

$$\text{LOD} = 3\frac{\partial}{\alpha} \tag{3}$$

$$\text{LOQ} = 3 \times \text{LOD} \tag{4}$$

### 2.7.3. Precision

The precision of the method was evaluated based on the study of repeatability and intermediate accuracy. Repeatability was verified for the analytical curve by analyzing for 3 concentration levels the relative standard deviation (RSD%), calculated according to equation below [38,39].

$$\text{RSD\%} = \frac{\sigma}{\overline{X}} \times 100 \tag{5}$$

where $\sigma$ and $\mu$ are the standard deviation and $\overline{X}$ is the average value.

The intermediate precision was evaluated by analysis of variances (ANOVA), *t*-test, and F-test of analytical curves found on different days and with different electrodes.

### 2.7.4. Recovery/Accuracy

Recovery was studied by applying 3 levels of analytical concentration, and accuracy was evaluated Equation (6) below.

$$R(\%) = \frac{\text{Cm}}{\text{Cex}} \times 100 \tag{6}$$

where Cm and Cex are the measured and expected concentrations, respectively.

*2.8. Stability*

The electrode stability study was conducted through successive glycerol analyses, where the area of the peak oxidation of the analyte was evaluated along with the analysis



until there was signal loss. The conditions of analysis of the study involved a working range of $-0.8$ to 0.70 V with 0.050 V s$^{-1}$ in 0.030 mol L$^{-1}$ KOH containing 1.80 mol L$^{-1}$ glycerol, using cyclic voltammetry as the analysis technique.

### 2.9. Glycerol Extraction in Biodiesel Sample

Glycerol was extracted from biodiesel following the following protocol: 800 µL of biodiesel, 800 µL milli-Q water, 800 µL ethanol HPLC grade, and 1600 µL of heptane were mixed in a glass vial and stirred on the vortex mixer for 2 min. Then for phase separation, the solution was centrifuged at 1500 RFC for 5 min. After this procedure, the aqueous phase was aliquoted and dried by nitrogen, and then resuspended in KOH 0.10 mol L$^{-1}$ to be able to be analyzed electrochemically.

## 3. Results and Discussion

### 3.1. Characterization

3.1.1. Characterization of Graphene-Based Material

To understand the effect and efficiency of the reduction of graphene oxide in the electrode, the active electrochemical area and heterogeneous electron transfer (HET) rate constant ($k^0$) were studied; these results can be seen in Table 1.

**Table 1.** Study of the electrochemically active area and $k^0$.

| Electrode | Area (cm$^2$) | $k^0$ (cm s$^{-1}$) |
|:---:|:---:|:---:|
| GO/GCE | $1.47 \times 10^{-2}$ | $4.72 \times 10^{-3}$ |
| rGO/GCE | $6.90 \times 10^{-2}$ | $1.32 \times 10^{-2}$ |

In the Table 1, it is clear that the reduction of graphene oxide was a crucial step for producing the electrode. After the electrochemical reduction of the GO, the electroactive area of the electrode increased by 4.5 times and the $k^0$ approximately 3 times.

In Figure 1, it is possible to notice that after the modification of the GCE with rGO, there was an increase in both the faradaic and the capacitive current, indicating electrode modification [20]. Using the Randles–Sevcik [28–30] equation (Equation (1) it was possible to calculate the electroactive area of the electrodes, showing that after GCE modification the electroactive surface area (ECSA) increased 64% (ECSA of GCE = 0.042 cm$^2$, ECSA of rGO/GCE = 0.069 cm$^2$).

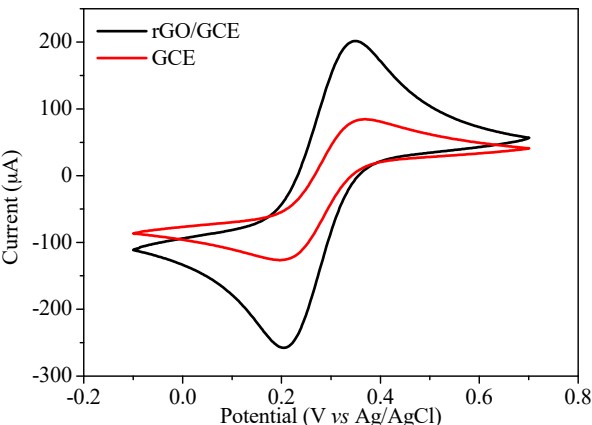

**Figure 1.** Voltammetric profile of the ferri-ferrocyanide couple at 20 mmol L$^{-1}$ using GCE and the rGO/GCE electrode with a 0.10 V s$^{-1}$ scan rate.

3.1.2. Characterization of Electrode Surface

SEM (Figure 2a–d) was used to investigate the distribution of particles in the electrode and TEM (Figure 2e,f) to investigate the morphologies and structures of the produced nanoparticles.

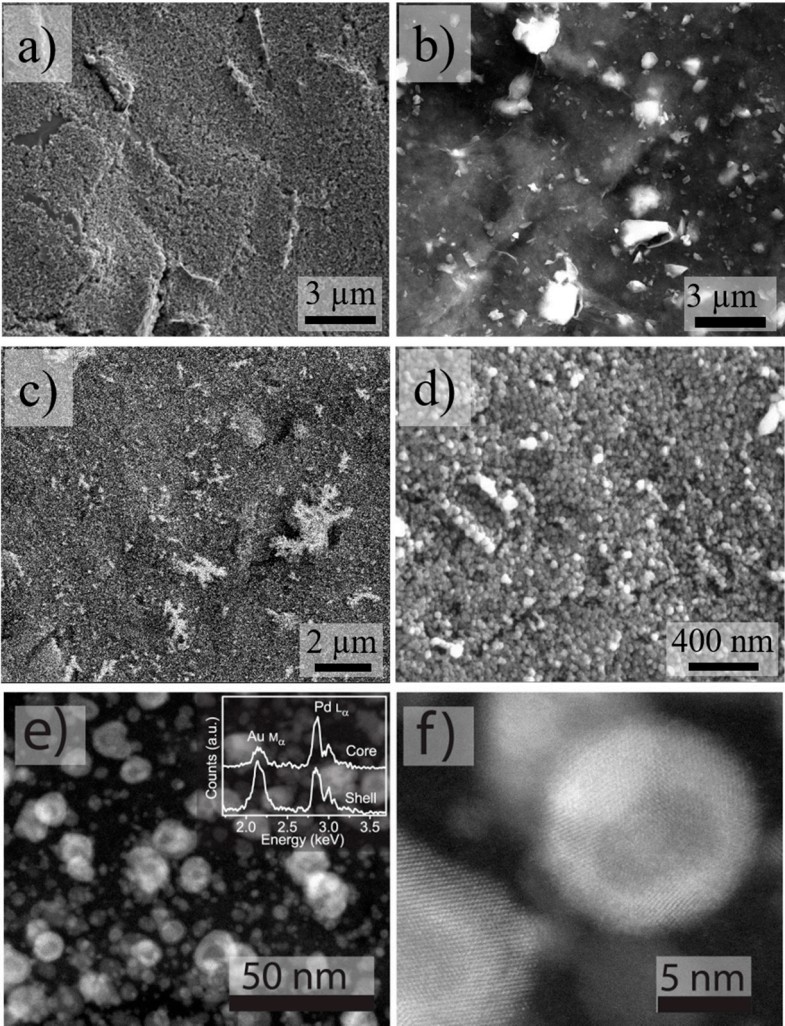

**Figure 2.** SEM (**a**–**d**) and TEM images (**e**,**f**) of the electrode revealing details of the fabrication steps. (**a**) Conventional SPE electrode. (**b**) SPE modified with rGO. (**c**) and (**d**) the electrode with core-shell gold–palladium nanoparticles. (**e**) STEM image showing the core-shell gold–palladium nanoparticles. The EDS elemental analysis is shown in the inset. (**f**) Details of the core-shell nanoparticles using high-resolution STEM (HR-STEM).

In Figure 2a–d is possible to note the different electrode surface morphologies after each modification step. The SPE electrode surface without any modification (Figure 2a) presented a smooth structure with some grooves and deformations characteristic of the electrode [40]. With rGO (Figure 2b) different reliefs and roughnesses are observed, characteristic of the overlapping of rGO sheets on electrode surface. Figure 2c,d show the electrode modified with core-shell gold–palladium nanoparticles with low and high magnification, respectively. With this analysis it was not possible to observe any morphological characteristics of the electrode, only that the particles had a small size. We moved to higher resolution, analyzing the nanoparticle using a TEM (Figure 2e,f). Figure 2e shows that there were abundant nanoparticles distributed across the whole section with a diameter of less than 20 nm. The EDS analysis (Figure 2e) and the HR-STEM (Figure 2f) of the particles show the formation of core-shell particles; the composition is of a nucleus rich in palladium and an external part rich in gold.

### 3.2. Electrochemical Characterization of the Electrode

In Figure 3 it is possible to observe the voltammetric profile of the Au@Pd/rGO/GCE electrode in an alkaline medium.

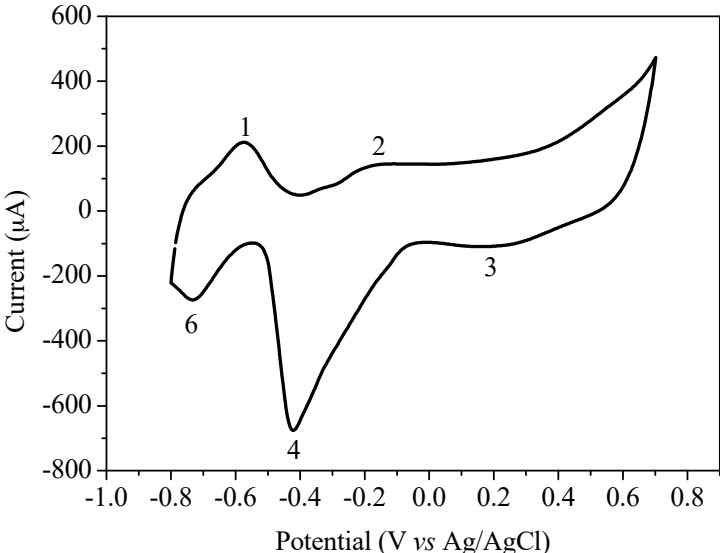

**Figure 3.** Voltammetric profile of the Au@Pd/rGO/GCE electrode in KOH 0.030 mol L$^{-1}$.

Observing the voltametric profile, it is possible to verify the presence of some peaks. Peaks 1 and 6 refer to the hydrogen adsorption and desorption, respectively. Peak 2 is related to OH$^{-}$ adsorption, and cathodic peaks 3 and 4 are related to the reduction of gold oxide particles and the reduction of AuPd alloys, respectively [25,41,42].

### 3.3. Partial Validation

### 3.3.1. Linearity

Figure 4 presents the cyclic voltammograms obtained using Au@Pd/rGO/GCE in a 0.030 mol L$^{-1}$ KOH at 50 mV s$^{-1}$ scan rate for different glycerol concentrations. It is possible to verify that the anodic peak current found at around $-0.20$ V (vs. Ag/AgCl) increases with the glycerol concentration.

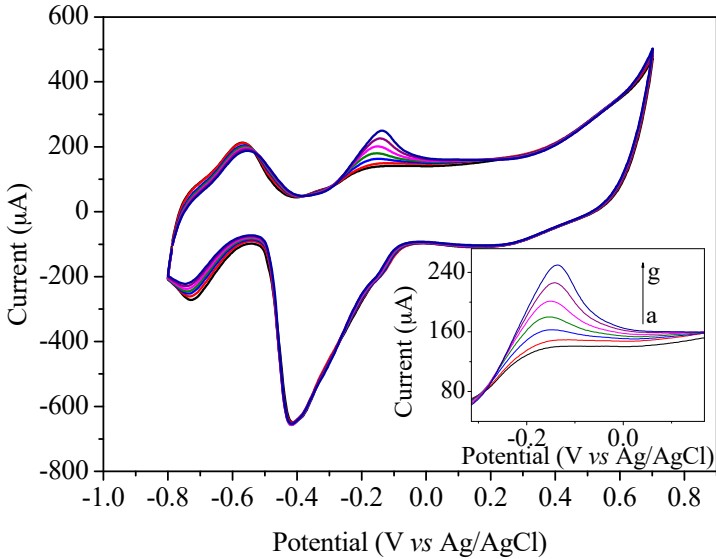

**Figure 4.** The cyclic voltammograms for different glycerol concentrations using Au@Pd/rGO/GCE in 0.030 mol L$^{-1}$ KOH at a 50 mV s$^{-1}$ scan rate. Voltammograms a–g are related to the glycerol concentrations at 0, 18.2, 36.5, 54.7, 73, 91.2, and 109 µmol L$^{-1}$, respectively.

Figure 5 shows the relationship between the peak area and the concentration of glycerol added to the system in the range of 18.2 and 109 µmol L$^{-1}$. The Pearson correlation

coefficient obtained for the linear curve was 0.9895, greater than 0.9000, thus showing that there is a strong linear relationship between the parameters (Figure 5) [35]. For the Cochran and Anderson–Darling tests, *p*-values equal to 0.9439 and 0.9352 were obtained at a significance level of 5%, respectively. Thus, we do not reject the hypothesis that the distribution of waste is normal and we do not reject the hypothesis of equality of variances, which is a normal and homoscedastic model [36,37,43].

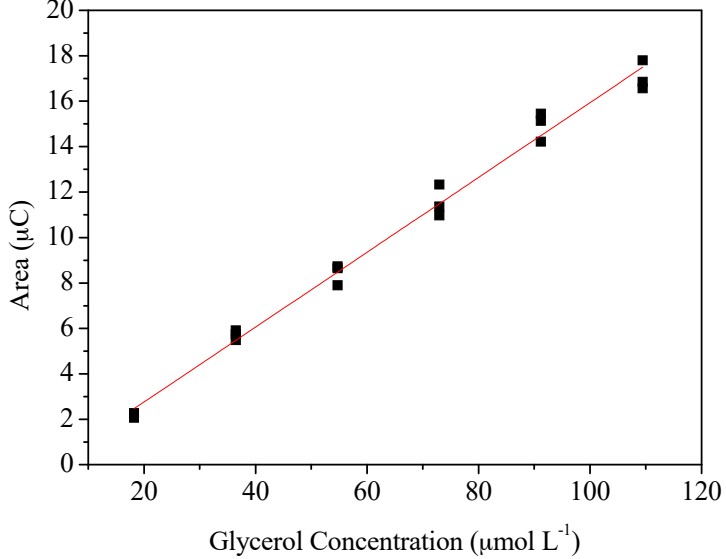

**Figure 5.** The analytical curve for glycerol in 0.030 mol $L^{-1}$ KOH (glycerol oxidation peak area versus glycerol concentration in triplicate) using the Au@Pd/rGO/GCE electrode.

### 3.3.2. Limit of Detection (LOD) and Quantification (LOQ)

According to Equations (3) and (4), the limits of detection and quantification of the electrode were 5.32 and 17.5 $\mu$mol $L^{-1}$, respectively.

### 3.3.3. Intermediate Precision

The intermediate precision of the method was assessed by studying the equality of the analytical curves from different days with different electrodes (Figure 6); for this the F test and the T test were used. For the evaluated curves, we obtained a $F_{cal}$ value equal to 1.80, less than the $F_{crit}$ of 2.27, and a $t_{cal}$ value equal to 1.70, less than the $t_{crit}$ of 2.03. These results indicate that the curves are equal to the 5% significance level.

### 3.3.4. Repeatability and Recovery

For the study of method repeatability and recovery, the same concentrations of the analytical curve were evaluated. For repeatability, the relative standard deviation (RSD) of the triplicate analyzes was evaluated (Equation (4)) and for recovery, the measured concentration (Cm) was evaluated in relation to the expected concentration (Cex) (Equation (5)), as can be seen in Table 2.

As can be seen from the results presented in Table 1, the method presented a maximum RSD of 4.73% and recovery between 90.9% and 103%; these results are within the maximum allowed value according to the INMETRO [38], where the maximum acceptable RSD is 20% and recovery must be in the range of 70% and 120%. These results show that the method is accurate and together with the intermediate precision, the repeatability data show that the method is also precise.

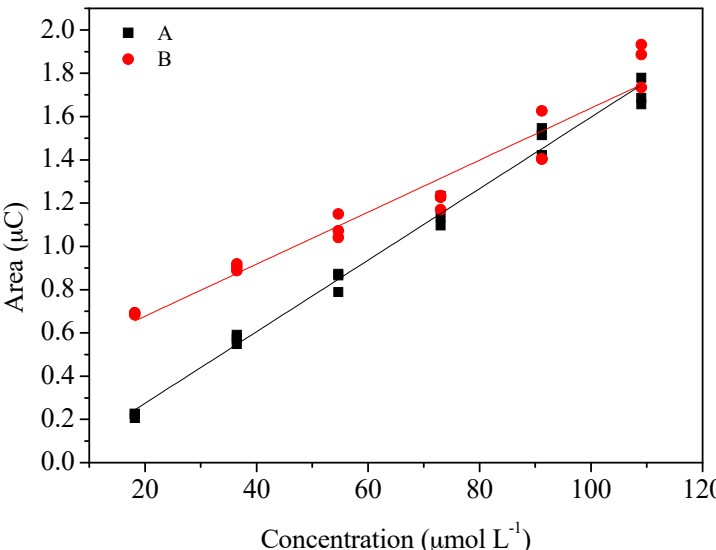

**Figure 6.** Comparison of 2 linear curves for glycerol analyses performed on different days with different electrodes.

**Table 2.** Study of method repeatability and recovery.

| Expected Concentration ($\mu$mol L$^{-1}$) | Measured Concentration ($\mu$mol L$^{-1}$) | RSD (%) | Recovery (%) |
|---|---|---|---|
| 18.2 | 16.5 | 3.26 | 90.9 |
| 36.5 | 38.0 | 2.81 | 104 |
| 54.7 | 54.4 | 4.21 | 99.4 |
| 73.0 | 73.3 | 4.73 | 101 |
| 91.2 | 93.9 | 3.39 | 103 |
| 109 | 107 | 2.98 | 98.0 |

### 3.4. Stability

The stability study was conducted by subjecting the electrode to analysis of 1.40 mmol L$^{-1}$ glycerol for 1000 cycles, using a 0.050 V s$^{-1}$ scan rate and a potential range from $-0.80$ to 0.70 V (Figure 7). The peak area of the glycerol oxidation was evaluated every 100 cycles. During the analysis, an RSD of 6.61% was obtained, lower than that allowed by IN-METRO [44] (20%), and up to 900 cycles there was no drop in signal for glycerol analysis. On comparing the first analysis with the last, there was a drop of only 3%. These results show that the electrode has excellent stability, losing the signal after 900 cycles.

### 3.5. Glycerol Analysis on Biodiesel

For the glycerol analysis, two extractions were performed and analyzed in different days with different electrodes. The samples were analyzed electrochemically by the standard addition method and compared with the values obtained by the spectrophotometric enzymatic method, a method that has been validated and patented by the laboratory (FR1872032). By the electrochemical method, concentrations of 0.0101% and 0.00998% $w/w$ were obtained, while for the same sample using the spectrophotometric enzymatic method a concentration of 0.0116% $w/w$ was obtained, thus resulting in recovery values of 87% and 86% $w/w$ for the analyzed samples. These results show that the proposed method can accurate and precisely determine glycerol levels in biodiesel samples using liquid–liquid extraction.

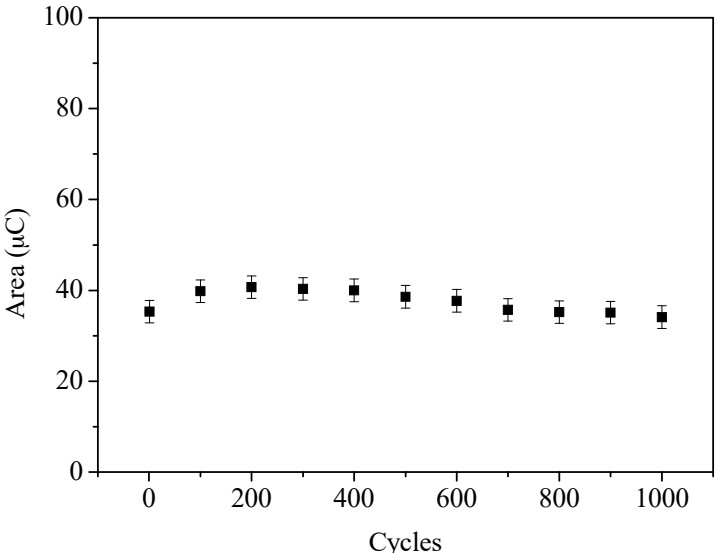

**Figure 7.** Study of electrode stability evaluating the glycerol oxidation peak every 100 cycles for 1000 cycles using a 1.40 mmol L$^{-1}$ glycerol solution.

## 4. Comparison with Literature

According to ASTM 6584-00 (United States of America), DINV51606 (Germany), EN 14105:2003 (Europe), and GB/T20828-2007 (China), and the maximum free glycerol concentration allowed in biodiesel is 0.02% *w/w* (2.17 mmol L$^{-1}$). All the techniques presented in Table 3 satisfy the established norms; however, the method developed in this work, despite not having the lowest LOQ, has the advantage of presenting an easy-to-manufacture electrode with relatively low cost, without the need for pretreatment of the sample (derivatization). The method provides quick results and less reagent consumption. The electrode has excellent stability, which, from an industrial point of view, is a desirable feature.

**Table 3.** Comparison of the proposed electrode with others in the literature.

| Sample Preparation | Electrode | Pretreatment | Sample Volume (g) | Technique | LOD (μmol L$^{-1}$) | LOQ (μmol L$^{-1}$) | Reference |
|---|---|---|---|---|---|---|---|
| SPE | - | No | 0.132 | Enzymatic-spectrophotometric | 0.836 | 2.71 | [44] |
| LLE | - | No | 1 | MCFA-SP | 10.9 | - | [45] |
| LLE | - | Yes | - | Spectrofluorimetric | 4.02 | - | [46] |
| LLE | CuONP/Pe(MWCNT)/GC | No | 5 | CA | 0.0631 | 0.210 | [8] |
| LLE | Au/SiO$_2$ cavity/ITO | No | 1 | DPV | 1.49 | 4.92 | [5] |
| LLE | Ni/CG | No | 50 mL | DPV | 33.0 | 108.9 | [2] |
| | PtRuSPUME | No | - | CA | 5.50 | 18.2 | [17] |
| LLE | Pt disk | No | 2 | CV | 25.0 | 82.5 | [47] |
| LLE | Au@Pd/rGO/GC | No | 0.7 | CV | 5.30 | 17.6 | This Work |

LLE: liquid–liquid extraction; SPE: solid phase extraction; CA: chronoamperometry; DPV: differential pulse voltammetry; CV: cyclic voltammetry; MCFA: multicommutation in flow analysis; SP: spectrophotometry.

## 5. Conclusions

The Au@Pd/rGO/GCE electrode showed low detection and quantification limits (5.3 and 17.5 μmol L$^{-1}$, respectively), with good repeatability and excellent stability (not losing efficiency after 900 cycles). With the proposed method it was possible to analyze glycerol in biodiesel samples using the LLE method, with recovery values of 86% and 87%. The results obtained in this study show that the formed nanocomposite film has excellent electrocatalytic activity for glycerol oxidation, and is therefore a good alternative as a glycerol electrochemical sensor in biodiesel samples and/or for other alcohols in different matrices.

**Author Contributions:** Conceptualization, V.M.P., E.S.R. and E.D.; Data curation, V.M.P. and B.S.A.; Formal analysis, V.M.P., K.L.d.S.C.A., B.S.A., D.R.F. and C.A.S.; Investigation, V.M.P., B.S.A. and D.R.F.; Methodology, V.M.P., K.L.d.S.C.A. and E.D.; Project administration, E.D.; Supervision, E.D. and C.A.A.; Validation, V.M.P.; Visualization, V.M.P.; Writing—original draft, V.M.P.; Writing—review & editing, E.S.R., C.A.A. and E.D. All authors have read and agreed to the published version of the manuscript.

**Funding:** This research received no external funding.

**Institutional Review Board Statement:** Not applicable.

**Informed Consent Statement:** Not applicable.

**Data Availability Statement:** Not applicable.

**Conflicts of Interest:** The authors declare no conflict of interest.

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
