# Peer review of "Electrochemical Analysis of Free Glycerol in Biodiesel Using Reduced Graphene Oxide and Gold/Palladium Core-Shell Nanoparticles Modified Glassy Carbon Electrode"

_processes, doi:10.3390/pr9081389_

Round 1

Reviewer 1 Report

Manuscript entitled "Electrochemical analysis of glycerol in biodiesel using reduced graphene oxide and gold/palladium core-shell nanoparticles modified glassy carbon electrode" reports modification of glassy carbon electrode with reduced graphene oxide and core-shell gold@palladium nanoparticles and use of this modified electrode for determination of glycerol in biodiesel.

The abstract is unnecessarily detailed, a description of the procedure for modifying the electrode is not necessary. The description of sample preparation is important, but should be simplified for easier understanding. On the contrary, the abstract also completely lacks what makes the presented work so fundamental and exceptional that it was worth writing this manuscript. In other words, the authors must fundamentally emphasize the novelty. Moreover there is no info on why it is important to determine glycerol in biodiesel - which is essential information.

Lines 34-37 are unnecessarily extensive, a maximum of one sentence included in the first paragraph is more than enough.

For methods mentioned on Lines 48-49, indicate which methods are used (accepted for glycerol control) and recognized for the determination of glyderol.

Process the information on lines 50-51 into a clear table, as usual - this is actually related to the previous reviewer request, so authors can summarize which techniques are certified and which are not in this table too. Authors present a new and promising method, so they must make a comparison with the current state of knowledge, so it is necessary to state any limitations, of course LOD, LOQ, LDR. Please also state what the requirements are for sample preparation before measurement! Methods that do not require extraction and allow direct measurement are certainly much more interesting!

On Line 85, please add water resistivity. In addition, you mention "milli-Q water" on page 4. So what water was used?

If you do not specify the rotor radius, the information "... the solution was centrifuged for 5 minutes at 3000 rpm." is absolutely useless. Please specify RCF.

The rGO used is of very poor quality and contains impurities, which can be expected from the mentioned method of GO preparation and which is well shown, inter alia, in Figure 2b. It is a pity that the authors did not use a higher quality product to modify the electrode. In addition, the use of such a product complicates the repetition of the study. After all, the process of graphene oxide reduction itself is not well documented and should be included in the Supplementary - just because it is an important step (or better say one of crucial step) for electrode modification.

The GO used is the biggest weakness of the study, which by the way can call it into question. It is recommended to purchase GO in high quality and to use this material to modify the electrode. The fact that the material prepared by the authors was published somewhere does not change anything. Most published GOs are in fact very questionable and of a very low quality!

Why didn't the authors transfer the entire determination to the SPE? The article would certainly be much more attractive in terms of practical application.

Minor comments (especially of a typographic nature):

  • please use the typographically correct symbol for minus and negative numbers (including exponents). The Hyphen symbol (-) is used in the manuscript instead of the correct Minus Sign.
  • please write chemical formulas in correct way. For complex compounds square brackets should be used, HAuCl4 should be better written as H[AuCl4].
  • please note that the company Sigma-Aldrich is now called MERCK!
  • please keep a consistent number of decimal places or valid digits, e.g. in record "... 5.0 V s-1 scan rate in the range of -1 to 1 V" (Line 102, Page 3) or "... range of -0.8 to 0.8 V" (Line 106, Page 3). The same stands for Table 1. Please check the entire document carefully and correct this recurring deficiency.
  • please use italic for variables, e.g. "C", "D" (Line 118, Page 3).
  • number "6.2 x 10-6 cm2 s-1" should be written with "Multiple SIgn" instead of "x" letter. Please check the entire document carefully and correct this recurring deficiency.

The similarity index is 21 % (according to Crossref Similarity Check). A document with marked matches is attached. It is recommended to rephrase some identical longer passages.

Author Response

  1. The abstract is unnecessarily detailed, a description of the procedure for modifying the electrode is not necessary. The description of sample preparation is important, but should be simplified for easier understanding. On the contrary, the abstract also completely lacks what makes the presented work so fundamental and exceptional that it was worth writing this manuscript. In other words, the authors must fundamentally emphasize the novelty. Moreover, there is no info on why it is important to determine glycerol in biodiesel - which is essential information.

Answer: The abstract was rewritten as suggested by the referee.

  1. Lines 34-37 are unnecessarily extensive, a maximum of one sentence included in the first paragraph is more than enough.

Answer: The first and second paragraph were revised as suggested by the referee.

  1. For methods mentioned on Lines 48-49, indicate which methods are used (accepted for glycerol control) and recognized for the determination of glycerol.

Answer: The standard method was recognized in the text.

  1. Process the information on lines 50-51 into a clear table, as usual - this is actually related to the previous reviewer request, so authors can summarize which techniques are certified and which are not in this table too. Authors present a new and promising method, so they must make a comparison with the current state of knowledge, so it is necessary to state any limitations, of course LOD, LOQ, LDR. Please also state what the requirements are for sample preparation before measurement! Methods that do not require extraction and allow direct measurement are certainly much more interesting!

Answer: In the topic 4 was added more comparison with other techniques.

  1. On Line 85, please add water resistivity. In addition, you mention "milli-Q water" on page 4. So what water was used?

Answer: The resistivity was including on the manuscript.

  1. If you do not specify the rotor radius, the information "... the solution was centrifuged for 5 minutes at 3000 rpm." is absolutely useless. Please specify RCF.

Answer: The velocity was changed for RCF.

  1. The rGO used is of very poor quality and contains impurities, which can be expected from the mentioned method of GO preparation and which is well shown, inter alia, in Figure 2b. It is a pity that the authors did not use a higher quality product to modify the electrode. In addition, the use of such a product complicates the repetition of the study. After all, the process of graphene oxide reduction itself is not well documented and should be included in the Supplementary - just because it is an important step (or better say one of crucial step) for electrode modification.

Answer: The revisor is right, the reduction of the graphene oxide is very important for the efficiency of the electrode. In the manuscript was include an electrochemical study of the electrode with reduced graphene oxide and graphene oxide.

The GO used is the biggest weakness of the study, which by the way can call it into question. It is recommended to purchase GO in high quality and to use this material to modify the electrode. The fact that the material prepared by the authors was published somewhere does not change anything. Most published GOs are in fact very questionable and of a very low quality!

Answer: The review is right. In fact, using high quality commercial GO would be much better, however unfortunately in Brazil research is not valued and we are increasingly lacking funding to develop our research. So, to use GO we made a collaboration in which graphite oxide was donated.

  1. Why didn't the authors transfer the entire determination to the SPE? The article would certainly be much more attractive in terms of practical application.

Answer: The study was not carried out on the SPE as we thought liquid-liquid extraction would be simpler and more available. The vacuum pump used in our laboratory for SPE is defective.

Minor comments (especially of a typographic nature):

  • please use the typographically correct symbol for minus and negative numbers (including exponents). The Hyphen symbol (-) is used in the manuscript instead of the correct Minus Sign.

Answer: The Hyphen symbol (-) was replaced by the correct Minus Sign.

  • please write chemical formulas in correct way. For complex compounds square brackets should be used, HAuCl4 should be better written as H[AuCl4].

Answer: The chemical formula of the H[AuCl4] was corrected.

  • please note that the company Sigma-Aldrich is now called MERCK!

Answer: The name of the company was corrected.

  • please keep a consistent number of decimal places or valid digits, e.g. in record "... 5.0 V s-1 scan rate in the range of -1 to 1 V" (Line 102, Page 3) or "... range of -0.8 to 0.8 V" (Line 106, Page 3). The same stands for Table 1. Please check the entire document carefully and correct this recurring deficiency.

Answer: The number of decimal places were checked and corrected.

  • please use italic for variables, e.g. "C", "D" (Line 118, Page 3).

Answer: The variables ‘’C’’ and ‘’D’’ were placed in italic format.

  • number "6.2 x 10-6 cm2 s-1" should be written with "Multiple SIgn" instead of "x" letter. Please check the entire document carefully and correct this recurring deficiency.

Answer: The letter ‘’x’’ was changed for the "Multiple SIgn"

The similarity index is 21 % (according to Crossref Similarity Check). A document with marked matches is attached. It is recommended to rephrase some identical longer passages.

Answer: Some identical longer passages was changed.

Reviewer 2 Report

The presented manuscript describes an electrochemical method for the glycerol determination in biodiesel on modified glassy carbon electrode. In my opinion, this work should be published in Processes after minor revision.

My comments and questions:

  1. Manuscript should be proof-read by someone who is proficient in English language. The language used in the manuscript is understandable, but very poor.
  2. Verse 82: purchased by instead of from.
  3. Purity of the auxiliary electrode is missing.
  4. How the authors prepare the surface of the GCE electrode before modification?
  5. The size of fonts and axis descriptions in the figures is different. Please, unify that.
  6. The abbreviation VS in the figures is not in italics.
  7. Did the authors try to employ another voltammetric technique? It is generally known that square-wave voltammetry or differential-pulse voltammetry are more sensitive. If not, why?

Author Response

  1. Manuscript should be proof-read by someone who is proficient in English language. The language used in the manuscript is understandable, but very poor.

Answer: The English language was revised by a professional.

2. Verse 82: purchased by instead of from.

Answer: We include the word purchased.

3. Purity of the auxiliary electrode is missing.

Answer: The purity was added in the manuscript.

4. How the authors prepare the surface of the GCE electrode before modification?

Answer: Before to modification of the GCE with reduced graphene oxide (rGO), the GCE was cleaned following the following steps, first the electrode was polished with alumina powder suspension (0.3 um) followed by a sonication in 3 mmol L-1 of HNO3 for 10 minutes and finally submitted to electrochemical cleaning using 0.5 mol L-1 sulfuric acid using the cyclic voltammetry technique at a scan rate of 1 V s-1 in a potential range of -1.0 to 1.0 V.

This paragraph was added in the topic 2.3.

5. The size of fonts and axis descriptions in the figures is different. Please, unify that.

Answer: The size of the fonts was corrected.

6. The abbreviation VS in the figures is not in italics.

Answer: The abbreviation VS was placed in italics.

7. Did the authors try to employ another voltammetric technique? It is generally known that square-wave voltammetry or differential-pulse voltammetry are more sensitive. If not, why?

Answer: The techniques of SWV and DPV are in fact much better for quantification, however for the analysis of alcohols it is important to regenerate the active sites of the metal hydroxides, that’s why we chose cyclic voltammetry because at the same time it is possible to quantify the analyte and regenerating the electrode surface. Without regeneration of the active sites, the electrode loses its efficiency for quantification. The cyclic voltammetry technique was chosen to regenerate the active sites of electrode.

Round 2

Reviewer 1 Report

The authors have done a number of changes in manuscript, responded to most of the reviewer's comments and questions, and the quality of the manuscript was improved significantly. Despite shortcomings regarding the rGO used, the study can be considered beneficial and interesting. In order not to limit the team's research next time, the reviewer offers the author's team a high-quality GO prepared in his laboratory for further work and possible potential cooperation. The sample will be provided free of charge. The authors can request contact to reviewer from editor if they are interested.